# Identification of Biological Functions and Prognostic Value of NNMT in Oral Squamous Cell Carcinoma

**DOI:** 10.3390/biom12101487

**Published:** 2022-10-15

**Authors:** Weixian Zhang, Yue Jing, Shuai Wang, Yan Wu, Yawei Sun, Jia Zhuang, Xiaofeng Huang, Sheng Chen, Xiaoxin Zhang, Yuxian Song, Qingang Hu, Yanhong Ni

**Affiliations:** 1Central Laboratory of Stomatology, Nanjing Stomatological Hospital, Medical School of Nanjing University, Nanjing 210008, China; 2Department of Oral Pathology, Nanjing Stomatological Hospital, Medical School of Nanjing University, Nanjing 210008, China; 3Department of Oral and Maxillofacial Surgery, Nanjing Stomatological Hospital, Medical School of Nanjing University, Nanjing 210008, China

**Keywords:** Nicotinamide N-methyltransferase, oral squamous cell carcinoma, prognosis, metabolomics, nicotinamide, epithelial–mesenchymal transformation

## Abstract

Background: Nicotinamide N-methyltransferase (NNMT) is a metabolic enzyme that catalyzes the methylation of nicotinamide (NAM) to generate 1-methyl nicotinamide (MNAM). Although previous studies have shown that NNMT is frequently dysregulated to promote the onset and progression of many malignancies, its expression profile, prognostic value and function in oral squamous cell carcinoma (OSCC) are still unknown. Methods: We used untargeted metabolomics based on mass spectrometry to analyze potential metabolite differences between tumors and matched adjacent normal tissues in 40 OSCC patients. Immunohistochemistry (IHC) was used to analyze the NNMT expression profile in OSCC, and the diagnostic and prognostic values of NNMT were evaluated. Next, qPCR and Western blot were used to compare the expression of NNMT in five OSCC cell lines. Stable transfected cell lines were constructed, and functional experiments were carried out to elucidate the effects of NNMT on the proliferation and migration of OSCC cells. Finally, gene set enrichment analysis (GSEA) was performed using The Cancer Genome Atlas (TCGA) data to investigate the potential functional mechanisms of NNMT in OSCC. Results: We found that the nicotinamide metabolic pathway was abnormally activated in OSCC tumor tissues compared with normal tissues. NNMT was expressed ubiquitously in tumor cells (TCs) and fibroblast-like cells (FLCs) but was absent in tumor-infiltrating lymphocytes (TILs). OSCC patients with highly expressed NNMT in TCs had higher risk of lymph node metastasis and showed a worse pattern of invasion (POI). Moreover, patients with highly expressed NNMT were also susceptible to postoperative recurrence. Highly expressed NNMT can independently predict shorter disease-free survival and recurrence-free survival. Functionally, we demonstrated that the ectopic expression of NNMT promoted OSCC tumor cell proliferation and migration in vitro. Conversely, silencing exerted significantly opposite effects in vitro. In addition, GSEA showed that highly expressed NNMT was mainly enriched in the epithelial–mesenchymal transformation (EMT) pathway, which displayed a significant positive correlation with the six classic EMT markers. Conclusions: Our study uncovered that NNMT may be a critical regulator of EMT in OSCC and may serve as a prognostic biomarker for OSCC patients. These findings might provide novel insights for future research in NNMT-targeted OSCC metastasis and recurrence therapy.

## 1. Introduction

Head and neck squamous cell carcinoma (HNSCC) is the seventh most prevalent malignancy worldwide and accounts for 3% of all new cancer cases and 1.5% of all cancer deaths [1]. According to global cancer statistics 2020, more than 800,000 new cancers and 450,000 deaths were reported. Of these, oral squamous cell carcinoma (OSCC) originating in the alveolar ridge, buccal mucosa, floor of the mouth, palate, tongue and other parts of the oral cavity accounted for approximately 380,000 new cases and 180,000 deaths in 2020 [2,3]. Although considerable efforts have been made in the past decades, about one-third of OSCC patients still eventually develop life-threatening and untreatable recurrent OSCC [4]. The five-year survival rate of OSCC patients is less than 60% because of cancer metastasis, neoplasm recurrence and drug resistance [5,6,7].

The current mainstays of OSCC treatments includes surgery, chemotherapy, radiotherapy or a combination of these modalities depending on the extent of the disease and the patients’ co-morbidity factors [8]. Unfortunately, treatment is often accompanied by serious complications when these therapies are used. In addition, the treatments affect the life quality of the individuals, mainly because they impair appearance, eating and speech dysfunction [9]. The recent emergence of molecular targeted therapy, which can directly interfere with specific molecules to block cancer growth, metastasis and recurrence, has brought revolutionary progress in the treatment of malignant tumors [10]. However, despite the promising approval of some targeted therapeutic drugs, a large proportion of OSCC patients are still limited by poor clinical efficacy, underscoring the urgent need to identify novel biomarkers for diagnosis and to provide individualized treatment.

Nicotinamide N-methyltransferase (NNMT), as a cytoplasmic enzyme with a molecular weight of 29 kDa belonging to the N-methyltransferases family, was originally regarded as providing a straightforward mechanism for regulating nicotinamide (NAM) levels. NNMT catalyzes the methylation of NAM and structurally related compounds by using the universal methyl donor S-Adenosyl methionine (SAM) to produce S-adenosyl-L-homocysteine (SAH) and 1-methylnicotinamide (MNAM) [11]. The catalytic product of NNMT, MNAM, can be further oxidized by aldehyde oxidase and excreted into the urine [12]. In recent years, NNMT has been reported to be involved in regulating multiple metabolic pathways by depleting methyl donors and producing active metabolites. NNMT is overexpressed in a variety of tumors and has been shown to promote progression and poor prognosis of several malignant tumors, such as gastric cancer, esophageal cancer and colorectal cancer [13,14,15]. For OSCC, Sartini D et al. reported that NNMT was highly expressed in OSCC patients and was negatively associated with lymph node metastasis [16]. However, this finding was incompatible with our expectations, and the sample size was small, with only about 20 cases.

Although previous study has shown that NNMT is frequently dysregulated to promote the onset and progression of many malignancies, including OSCC, its spatial expression profile, prognostic value and biological function in OSCC remain controversial, which is crucial for future studies of NNMT-targeted therapy for OSCC. In this study, we sought to detect the expression pattern of NNMT in OSCC, including tumor cells (TCs), fibroblast-like cells (FLCs) and tumor-infiltrating lymphocytes (TILs). We further evaluated the correlation between NNMT and clinicopathological features and their prognostic value. Moreover, stable transfected cell lines were constructed, and functional experiments were performed to elucidate the effects of NNMT on the proliferation and migration of OSCC cells. Finally, gene set enrichment analysis (GSEA) was performed using The Cancer Genome Atlas (TCGA) data to discover the HALLMARK pathway difference between high and low NNMT expression for the potential functional mechanisms.

## 2. Materials and Methods 

### 2.1. Patients and Samples

Tissue samples from 40 OSCC patients (each including tumor and adjacent normal tissue) were recruited to the study for untargeted analysis. All tissues were frozen in liquid nitrogen and stored at −80 °C until the sample was prepared for the assay. Samples and the follow-up data of 90 primary OSCC patients who had received cancer surgery from 2015 to 2017 were obtained from the pathology department of Nanjing Stomatological Hospital, Medical School of Nanjing University. The inclusion and exclusion criteria of patients were the same as those of our previous studies [17]. These patients were followed up for 3 to 71 months, and the median was 24 months. All patients provided informed consent for the collection of the study specimens. All the schemes of this study were approved by the Ethics Committee of Nanjing Stomatology Hospital. The study was conducted in accordance with the Declaration of Helsinki of the World Medical Association.

### 2.2. GC-MS Untargeted Analysis

Gas chromatography–mass spectrometry (GC-MS) untargeted analysis was used to find differences in potential metabolites between tumors and adjacent normal tissues in 40 OSCC patients. Untargeted analysis was performed by Trace 1310 Gas Chromato-graph equipped with an AS 1310 auto sampler, connected to a TSQ 8000 triple quadrupole mass spectrometer (Thermo Scientific, Waltham, MA, USA). Materials and specific steps for GC–MS analysis were based on the previously described method [18].

### 2.3. Cell Culture and Reagents

The human OSCC cell lines CAL33, CAL27, HSC3, HN6, OSCC3 and immortalized human oral keratinocyte (HOK) were kept in our laboratory and were maintained in the Dulbecco’s Modified Eagle Medium, high glucose (DMEM-H) supplemented with 10% fetal bovine serum and 1% penicillin–streptomycin. All cell lines were authenticated using Short Tandem Repeat (STR) analysis and cultured at 37 °C in a standard humidified atmosphere of 5% CO_2_. All cell culture reagents were obtained from Gibco (Thermo Fisher Scientific, Waltham, MA, USA).

### 2.4. RNA Extraction and Real-Time PCR Analyses

RNA was extracted using Trizol reagent by following the manufacturer’s procedure. The concentration and purity of the RNA were determined by measuring the absorbance at 260 and 280 nm using NanoDrop One (Thermo Fisher Scientific, Waltham, MA, USA). Total RNA was reversed into cDNA using HiScript III RT SuperMix (Vazyme Biotech Co., Ltd., Nanjing, China). The relevant expression of the genes was determined via AceQ^®^ qPCR SYBR^®^ Green Master Mix (Vazyme Biotech Co., Ltd., Nanjing, China.). The primer sequences used were as follows: forward primer 5-AGACAGGCGGTCAAGCAGGT-3 and reverse primer 5-AGACACAGTGTGCTGAGCACG-3 for human NNMT (NM_006169); forward primer 5-GGAGCGAGATCCCTCCAAAAT-3 and reverse primer 5-GGCTGTTGTCATACTTCTCATGG-3 for human GAPDH. All primer sequences were purchased from Invitrogen (USA). Gene expression was normalized to GAPDH and calculated using 2^−ΔΔCT^ method.

### 2.5. Western Blot

After cells in six-well plate reached a confluency of 80% to 90%, the cell lysates acquired by scraping the cultured cells were lysed in Radio-Immunoprecipitation Assay (RIPA) lysis buffer with a mixture of protease and phosphatase inhibitors on ice. Equal amounts of proteins were separated through sodium dodecyl sulfate polyacrylamide gel electrophoresis (SDS-PAGE) and blocked with 3% bovine serum albumin (BSA) for 1 h at room temperature. After incubation overnight with a primary antibody at 4 °C, with gentle shaking and horseradish peroxidase (HRP) conjugated secondary antibody for 1 h, protein bands were detected using the Tanon-5200 Chemiluminescent Imaging System (Tanon5200).

### 2.6. Plasmid Construction and Lentivirus Infection

We first performed the transient transfection of OSCC3 cells for 24, 36 and 48 h using small interfering RNA (siRNA) kits for NNMT, including three siRNAs and control. We then detected their NNMT mRNA expression after transfection by PCR, respectively, to screen the base sequence and transfection time with the highest transfection efficiency. After that, the siRNA with the highest transfection efficiency was cloned into pLVX-Puro, i.e., pLVX-shNNMT. The full length of the cDNA sequences of NNMT was cloned into lentivirus vector pCDH-CMV, i.e., pCDH-NNMT. Plasmids were transfected into 293T cells via Polyjet, and the supernatants were collected 2 days after transfection. After being filtered through a 0.45 μm PES filter, the viral supernatants were either directly added to OSCC3 or stored at −80 °C. Lentivirus particles overexpressing NNMT and controls were purchased from Genechem (Shanghai, China) and then utilized to transduce cells. Cell lines stably expressing NNMT were obtained after selection with puromycin (Sigma) for at least 1 week. The target sequences for NNMT of siRNAs were as follows: siRNA1, 5-GCAGAAAGCCAGATTCTTA-3; siRNA2, 5-CGTCGTCACTGACTACTCA-3; siRNA3, 5-CTTCCACCATGGCCAACAA-3. All siRNAs were purchased from RiboBio (Guangzhou, China). Transfection efficiency was confirmed by qPCR and Western blot.

### 2.7. Immunohistochemistry (IHC) and Quantification

Slides of formalin-fixed and paraffin-embedded tissues were deparaffinized with xylene and rehydrated in an ethanol series. Antigen retrieval was performed with 10 mM of citrate buffer (92 °C for 30 min) in a pressure cooker. The endogenous peroxidase activity was blocked with 5% BSA and then incubated overnight with primary antibody against NNMT (1:10,000 dilution; ab119758, Abcam) at 4 °C. Sections were incubated at 37 °C for 40 min using the Polink-2 plus HRP detection kit as a secondary antibody. Finally, slides were developed in diaminobenzidine (DAB). Gene expression was evaluated according to stain intensity and the percentage of positive cells. The staining intensity was scored as follows: 0, negative; 1, weak; 2, moderate; and 3, high intensity. The positive percentage was scored as follows: 0, no positive staining; 1, in between 1% and 25% cells; 2, in between 26% and 50% cells; 3, in between 51% and 75% cells; 4, in more than 75% cells. The final score was obtained by multiplying the two scores. All scorings were conducted by two pathologists without knowledge of the patients’ clinical characteristics or outcome. The expression of NNMT was defined as “low” when it was lower than the average value and as “high” when it was equal to or greater than the average.

### 2.8. CCK8 Assay

The constructed stable-transformation cell lines and control cells were seeded in 96-well plates at an initial seeding density of 3000 cells per well. The proliferation rate was determined after 0, 24, 48, 72 and 96 h of incubation. It was calculated according to the following formula: [(As−Ab)/(Ac−Ab)] × 100%, where Ab is the absorbance of the blank wells, Ac is the absorbance of the control wells, and As is the absorbance of the test wells.

### 2.9. Three-Dimensional Cell Culture Assay

The constructed stable-transformation cell lines and control cells were added in triplicate to low adsorption spherical culture plates (200 μL/well) at a concentration of 5 × 10^4^ cells/mL, respectively. The diameter of tumor spheres was observed daily for the following seven days.

### 2.10. Wound Healing Assay

The constructed stable-transformation cell lines and control cells were seeded in 6-well plates until 100% fusion. After overnight starvation with serum-free DF12 medium, the wounds were scored with a micropipette tip, and the cells were washed to remove dislodged cells and debris. The same area of the wound was photographed at 0 and 18 h to determine the wound closure of the cells.

### 2.11. Migration Assays

Migration assays were conducted according to the procedure described previously [17]. Briefly, experimental and control cells were added to the upper chamber of a Transwell (boyden chamber) (200 μL/well) at a concentration of 5 × 10^5^ cells/mL. The number of migrated cells at 24 h was qualified by the blinded counting of invaded cells on the lower surface of the membrane, with five fields per chamber, respectively.

### 2.12. Public Data Download and Processing

The gene expression data and clinical information of HNSCC patients in the TCGA database were downloaded using the R package TCGA biolinks, and non-OSCC patient samples were excluded. Paired differential expression analysis of OSCC primary tumors with adjacent normal tissues was then performed using the R package limma. GSEA enrichment analysis was performed using the R package cluster Profiler: First, the NNMT were classified into high and low expression groups according to their expression levels, and the log_2_FC of genes between the high and low expression groups was calculated using the R package limma. GSEA enrichment analysis was then performed using log_2_FC as the weight of the enrichment analysis. The gene sets were downloaded from the Hallmark gene set in the Msigdb database. Finally, the core genes obtained from the enrichment were mapped into a heat map using the R package complex heatmap.

### 2.13. Statistical Analysis

Statistical analyses and figure processing were performed using GraphPad Prism software 8.0 and SPSS software (Version 22.0). The paired t test was used to compare the mRNA and protein expression of NNMT between tumor tissues and adjacent normal tissues. The independent samples t test was used to analyze the difference between two groups. We evaluated the correlation between NNMT expression and clinicopathological characteristics of patients with OSCC using the Pearson’s chi-square test. We further analyzed whether increased NNMT was associated with postoperative recurrence using the Mann–Whitney U test. KM survival analysis was performed to analyze the survival of recruited patients for the prognostic significance of NNMT. Survival analysis including overall survival (OS), metastasis free survival (MFS), recurrence-free survival (RFS) and disease-free survival (DFS) were evaluated using the Kaplan–Meier (KM) and log-rank tests. The risk factors for OSCC were estimated using the univariate Cox proportional hazards regression model first. A multivariate Cox model was then constructed to estimate the adjusted hazard ratio (HR) and 95% confidence interval (CI) for NNMT expression. All the analyses were two-sided tests and were considered statistically significant at a *p* < 0.05.

## 3. Results

### 3.1. Aberrant Activation of Nicotinamide Metabolic Pathway in Tumor Tissues of OSCC Patients

In order to identify the metabolic differences between tumors and adjacent normal tissues, we performed GC-MS-based untargeted analysis. We found significantly higher NAM levels in the tumor tissue than in the adjacent normal tissue of OSCC patients, as well as significantly altered metabolites such as SAM and SAH (Figure 1a). The quantification of NAM, SAM and SAH in tumors was higher in tumor tissue than in normal tissue, respectively (Figure 1c–e, *p* < 0.0001). In this study, we focused on the metabolite changes in the NAM metabolic pathway and likewise on the conversion process of SAM to SAH in the methionine cycle. The above metabolite changes suggested that the NAM metabolic pathway was in an aberrantly activated state in tumor tissues of OSCC patients and that NNMT was the key metabolic enzyme catalyzing this process (Figure 1b).

### 3.2. NNMT Was Widely Expressed in the TCs and FLCs of OSCC Tumor Microenvironment

Data from TCGA demonstrated that the NNMT mRNA level was ubiquitously expressed in HNSCC patients (Figure 2a). Compared with the adjacent normal tissues, NNMT was significantly upregulated in tumor tissues of OSCC patients (*p* < 0.05, Figure 2b). Thus, to validate the expression of NNMT in OSCC, the current study included 90 patients with OSCC, as shown in Table 1. Typical IHC staining of NNMT low and high expression in TCs, FLCs and TILs was presented (Figure 2c). We found that NNMT was ubiquitously expressed in TCs and FLCs but was absent in TILs (Figure 2d).

### 3.3. Correlation between NNMT Expression and Clinicopathological Characteristics

We further evaluated the correlation between the expression of NNMT and clinicopathological features of OSCC patients, including gender, age, smoking habit, T stage, lymph node metastasis (LNM), differentiation, depth of invasion (DOI) and worst pattern of invasion (WPOI) (Table 1). Results showed that high expression of NNMT in TCs (NNMT^TCs^) was closely related to higher risk of lymph node metastasis (*p* < 0.01, Figure 3a) and WPOI (*p* < 0.01, Figure 3b). However, NNMT had no obvious correlation with gender, age, smoking habit, T stage or differentiation (all *p* > 0.05).

### 3.4. Upregulated NNMT^TCs^ Correlated with Post-Operative Recurrence and Poor Survival

Considering that upregulated NNMT has a malignant association with poor clinical outcomes, we further analyzed the relationship between NNMT expression with 5-year postoperative recurrence, as well as metastasis. Our results indicated that increased NNMT in OSCC had a significantly higher risk of postoperative recurrence after surgery but had no significant correlation with metastasis (*p* < 0.05, Figure 3c). To identify the prognostic value of NNMT, the Kaplan–Meier survival and log-rank tests were used, demonstrating that enhanced NNMT^TCs^ expression had shorter overall survival (Figure 4a), recurrence-free survival (Figure 4b) and disease-free survival (Figure 4d). However, metastasis-free survival showed no significant difference in NNMT^TCs^ (Figure 4c). Results from the analysis of OSCC patient data downloaded from the TCGA database further validated our conclusions (Figure 4e–h).

### 3.5. Expression Level of NNMT^TCs^ Was Independent Prognostic Factor for OSCC

Univariate and multivariate Cox regression analyses were used to analyze the prognostic value of distinct clinicopathological variables. Our results confirmed that gender, age, smoking habits, T stage, differentiation and chemoradiotherapy had no significant predictive value for OSCC. However, LNM, DOI, WPOI and high expression of NNMT^TCs^ were risk factors for OS, RFS, MFS and DFS in OSCC. Furthermore, multivariate analyses indicated that high expression of NNMT in TCs was an independent risk factor of RFS (Table 2) and DFS (Table 3) for OSCC.

### 3.6. Upregulated NNMT Promoted Proliferation and Migration of OSCC Cells

To investigate the function of NNMT in OSCC, we first compared the expression of NNMT in OSCC cell lines using qPCR (Figure 5a) and Western blot (Figure 5d). Among the five tumor cell lines, the expression of NNMT in OSCC3 was higher than human oral epithelial keratinocytes (HOK), while NNMT expression of the remaining four tumor cell lines was lower than HOK. Then, the NNMT overexpression vector or empty vector was stably transfected into HN6 with low NNMT expression and verified by qPCR (Figure 5b) and Western blot (Figure 5e). In order to uncover the function of NNMT in tumor cells, we first carried out the three-dimensional cell culture assay and wound healing assay and found that overexpression of NNMT in HN6 cells promoted cell proliferation (Figure 5g) and migration (Figure 5h). In addition, we also screened the optimal transfection conditions by transient transfection using siRNAs kit and found that siRNA2 had the highest transfection efficiency at the 36th hour after transfection (Appendix A). The lentivirus was then packaged with siRNA2 and pLVX-Puro to construct stable transfection cell lines. Conversely, we confirmed that NNMT knockdown in OSCC3 (Figure 5c,f) with high NNMT expression exerted opposite effects on cell proliferation (*p* < 0.001, Figure 5i) and migration (*p* < 0.01, Figure 5j) capacity by CCK8 assay and Transwell migration assay.

### 3.7. Signaling Pathways Involved in NNMT Expression in OSCC

In order to investigate the mechanism by which NNMT regulates OSCC proliferation and migration, we first excluded the non-OSCC sample in the HNSCC data (downloaded from the TCGA database), and then performed GSEA on these data to understand the molecular basis of the oncogenic property and identify the potential signaling pathways involved in NNMT expression. The epithelial–mesenchymal transition was the top signaling pathway most significantly enriched in the hallmark pathway (Figure 6a,b). Furthermore, we found that NNMT expression was significantly and positively correlated with the six EMT classical markers [19]: SNAI1, SNAI2, Twist1, VIM, ZEB1 and ZEB2 (all *p* < 0.01, Figure 6c). The relationship of NNMT and EMT gene sets is shown in Figure 6d. These findings demonstrated that NNMT functioned to regulate cell proliferation and migration of OSCC cells, perhaps acting through the EMT pathway (Figure 6e).

## 4. Discussion

As a small molecule methyltransferase in the human body responsible for the N-methylation of the nicotinamide, NNMT has been reported to promote proliferation or migration of several malignant tumors, including gastric cancer, esophageal cancer and OSCC [13,14,16]. Previous studies reported the high expression of NNMT in OSCC patients was negatively associated with lymph node metastasis. However, this was not relevant to our expectations, and their sample size was small, with only about 20 cases [16]. Additionally, little is known about the spatial expression profile and clinical significance of NNMT in different cell types of the tumor microenvironment (TME). In the present study, the spatial expression pattern of NNMT in different major cell types of TME was confirmed for the first time. Our results suggested that NNMT was ubiquitously expressed in TCs and FLCs of OSCC patients but was absent in TILs. Further analysis demonstrated that patients with highly expressed NNMT^TCs^ had a higher risk of lymph node metastasis and showed a worse WPOI. Moreover, patients with highly expressed NNMT in TCs were susceptible to postoperative recurrence. Although our results suggested that NNMT expression does not correlate with metastasis, this may have been due to the small number of patients with metastases in the 90 samples. The WPOI and DOI have been shown to be two valuable prognostic indicators of local recurrence in OSCC [20,21], which is also consistent with our findings. Those with highly expressed NNMT^TCs^ independently predicted shorter recurrence-free survival and disease-free survival.

We then investigated the function of NNMT in OSCC in vitro and indicated that the ectopic expression of NNMT in OSCC cells promoted cell proliferation and migration, while knockdown of NNMT had the opposite effect. Consistent with this, data from the transfection assay portion of one study showed that NNMT activity promoted cancer cell migration and invasion [22]. MTT analysis demonstrated that the silencing of NNMT significantly suppressed cell proliferation in GBM cancer cells [23]. Accumulating evidence suggests that EMT is associated with tumor initiation, invasion, metastasis, and resistance to therapy [24]. Enrichment analysis of OSCC data in the TCGA database also revealed that EMT was the most significantly enriched gene in the NNMT high expression group. Collectively, these results indicated that NNMT exerts oncogenic properties in OSCC by promoting cell proliferation and metastasis and could be a potential therapeutic target for OSCC.

We also found wide NNMT expression in FLCs (NNMT^FLCs^), which may suggest the importance of NNMT^FLCs^ in the progression and metastasis of OSCC. Eckert, M.A. et al. reported that stromal NNMT supported ovarian cancer migration, proliferation, and growth and metastasis [25]. FLCs are one of the most dominant cellular components in the mesenchyme; thus, the expression of NNMT in the FLCs should not be neglected. FLCs are known to be one of the most abundant stromal components in TME, which are capable of producing oncogenic extracellular matrix (ECM), secreting cytokines, participating in cytoskeletal rearrangement, and contracting collagen fibers [26]. Numerous studies have shown that FLCs have a prominent role in cancer pathogenesis, which has significant clinical implications [27]. It is becoming clear that analyzing the effect of NNMT expression in FLCs may help clarify the mechanism behind different responses to therapy and thus help determine possible clinically targeted therapeutic drugs for OSCC.

## 5. Conclusions

We demonstrated that NNMT is highly expressed in OSCC and can predict poor prognosis for OSCC patients, which might serve as a potential molecular diagnostic marker for OSCC. In vitro studies revealed that knockdown of NNMT effectively inhibited OSCC cell proliferation and migration, potentially acting through the EMT pathway. Therefore, beyond its currently discussed role as a molecular diagnostic marker, NNMT represents a promising new target for the treatment of OSCC and potentially other cancer entities [28]. In conclusion, our findings suggest that NNMT contributes to tumor progression and may serve as a promising therapeutic target for OSCC.

## Figures and Tables

**Figure 1 biomolecules-12-01487-f001:**
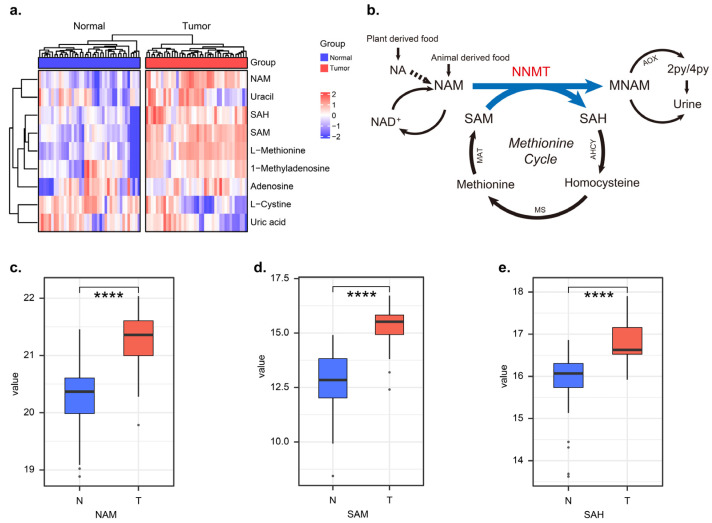
The nicotinamide metabolic pathway was in an abnormal activation state in OSCC tumor tissues. (**a**) Heatmap shows differential metabolites in tumor and adjacent normal tissues of OSCC, *n* = 40. (**b**) Nicotinamide N-methyltransferase (NNMT)-mediated methyl transfer from S-adenosyl-L-methionine (SAM) to nicotinamide (NAM), forming 1-methylnicoti-namide (MNA) and S-adenosyl-L-homocysteine (SAH). (**c**–**e**) The relative metabolic levels of NAM, SAM and SAH in tumors and normal tissues of OSCC patients, respectively, *n* = 40. **** represents that differences were considered statistically significant with *p* < 0.0001.

**Figure 2 biomolecules-12-01487-f002:**
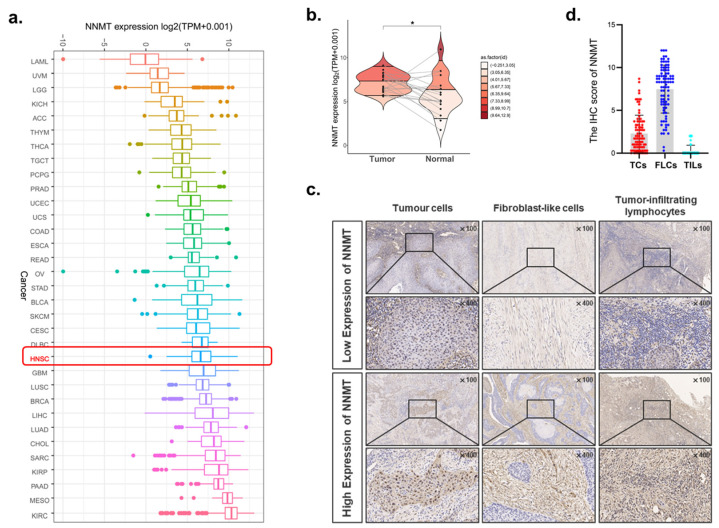
Expression of NNMT in OSCC and other cancers. (**a**) The NNMT expression in different tumor types using TCGA database. (**b**) RNA-seq data from TCGA study showed upregulation of NNMT in OSCC as compared with adjacent normal tissues (paired and unpaired samples, *n* = 19). (**c**) Typical IHC staining of NNMT high and low expression in TCs, FLCs and TILs, *n* = 90. (**d**) The IHC score of NNMT in TCs, FLCs and TILs from OSCC patients, *n* = 90. TCs, tumor cells; FLCs, fibroblast-like cells; TILs, tumor-infiltrating lymphocytes. * represents that differences were considered statistically significant with *p* < 0.05.

**Figure 3 biomolecules-12-01487-f003:**
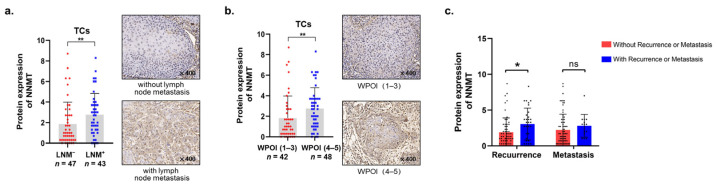
Correlation between the expression of NNMT and different clinical characteristics. (**a**,**b**) NNMT with different lymph node metastasis and WPOI in TCs. (**c**) Correlation between NNMT expression and the recurrence and metastasis status in TCs. * and ** represent differences that were considered statistically significant with *p* < 0.05 and *p* < 0.01, respectively; and ns represents no statistical differences.

**Figure 4 biomolecules-12-01487-f004:**
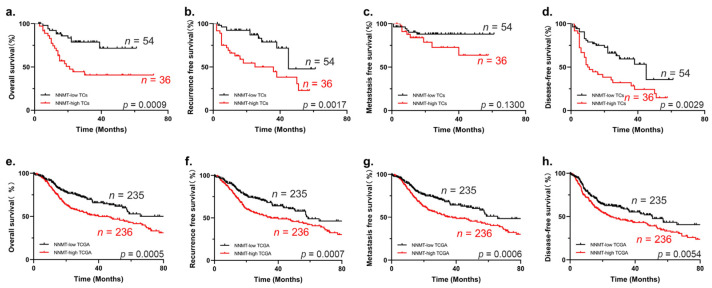
Kaplan–Meier survival curves for OS, RFS, MFS and DFS of OSCC patients. (**a**–**d**) High expression of NNMT has poor OS, RFS and DFS of patients according to our cohort, *n* = 90. (**e**–**h**) High expression of NNMT also has poor OS, RFS, MFS and DFS of patients using TCGA database. OS, overall survival; RFS, recurrence-free survival; MFS, metastasis-free survival; DFS, disease-free survival.

**Figure 5 biomolecules-12-01487-f005:**
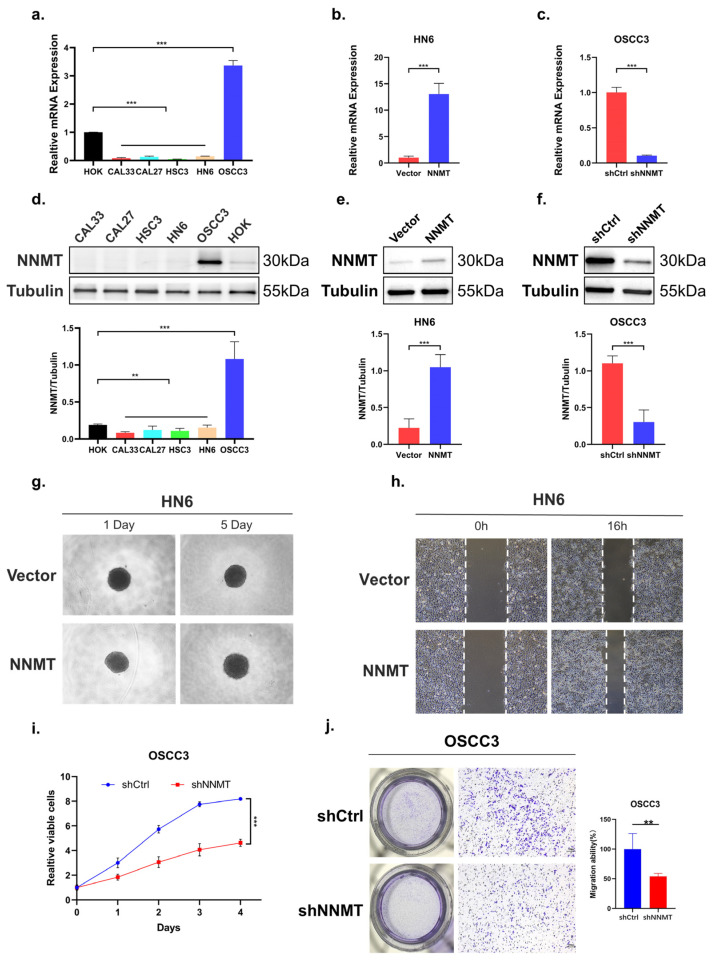
NNMT promoted the cell proliferation and migration of OSCC cells. (**a**,**d**) NNMT mRNA and protein expression in several OSCC cell lines. (**b**,**e**) Ectopic expression of NNMT in HN6 cells, confirmed using qPCR and Western blot. (**c**,**f**) Knockdown expression of NNMT in OSCC3 cells, confirmed using qPCR and Western blot. (**g**,**h**) Ectopic expression of NNMT significantly promoted proliferation and migration ability. (**i**,**j**) Knockdown expression of NNMT significantly inhibited proliferation and migration ability. ** and *** represented differences were considered statistically significant with *p* < 0.01 and *p* < 0.001, respectively.

**Figure 6 biomolecules-12-01487-f006:**
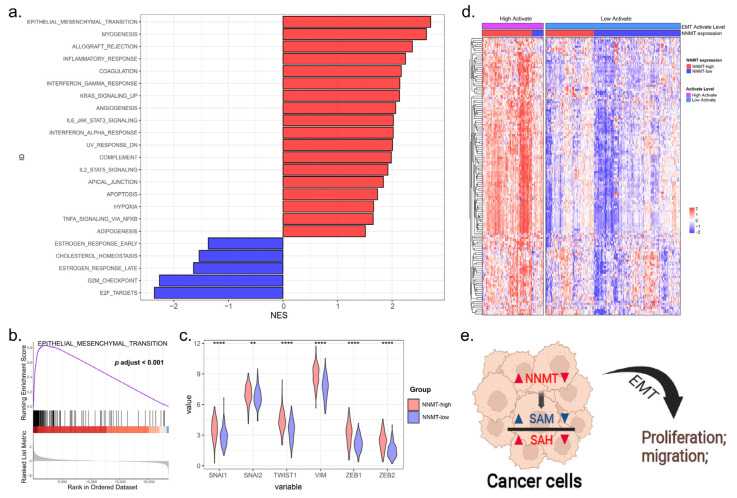
The expression of NNMT was related to the EMT pathway. (**a**) GSEA analysis for differently expression mRNAs in OSCC tissues with NNMT expression in hallmark datasets. (**b**) Significant upregulation of EMT gene set in OSCC tissues with NNMT-high expression. (**c**) The expression levels of the six classical markers of EMT in the NNMT-high group were significantly higher than those in the NNMT-low group. (**d**) Heatmap shows the relationship between EMT-related genes and NNMT expression levels in OSCC tissues. (**e**) NNMT promotes tumor cell proliferation and migration through the EMT pathway. EMT, epithelialؘ–mesenchymal transition; GSEA, Gene Set Enrichment Analysis. ** and **** represented differences were considered statistically significant with *p* < 0.01 and *p* < 0.001, respectively.

**Table 1 biomolecules-12-01487-t001:** Correlation between clinicopathology of OSCC patients and NNMT expression in TCs.

Clinical Variables	TCs		χ^2^	*p* Value
	Low *n* (%)	High *n* (%)		
**Gender**				
**Male**	27 (60.0)	18 (40.0)	0.000	1.000
**Female**	27 (60.0)	18 (40.0)		
**Age**				
**<60**	16 (55.2)	13 (44.8)	0.415	0.519
**≥60**	38 (62.3)	23 (37.7)		
**Smoking**				
**No**	40 (60.6)	26 (39.4)	0.038	0.846
**Yes**	14 (58.3)	10 (41.7)		
**T stage**				
**I–II**	34 (60.7)	22 (39.3)	0.032	0.859
**III–IV**	20 (58.8)	14 (41.2)		
**Lymph node metastasis**				
**No**	33 (70.2)	14 (29.8)	4.275	0.039*
**Yes**	21 (48.8)	22 (51.2)		
**Differentiation**				
**Well**	10 (62.5)	6 (37.5)	0.051	0.822
**Moderate/poor**	44 (59.5)	30 (40.5)		
**DOI**				
**<5 mm**	24 (64.9)	13 (35.1)	0.620	0.431
**≥5 mm**	30 (56.6)	23 (43.4)		
**WPOI**				
**I–III**	32 (72.7)	12 (27.3)	5.810	0.016 *
**IV–V**	22 (47.8)	24 (52.2)		

DOI, depth of invasion; WPOI, worst pattern of invasion, χ^2^, Pearson’s chi-squared test. * represents that differences were considered statistically significant with *p* < 0.05.

**Table 2 biomolecules-12-01487-t002:** Cox-regression analysis of OS and RFS in OSCC patients.

Variables	Univariate Analysis						Multivariate Analysis					
	Overall Survival			Recurrence Free Survival			Overall Survival			Recurrence Free Survival		
	HR	95%CI	*p* Value	HR	95%CI	*p* Value	HR	95%CI	*p* Value	HR	95%CI	*p* Value
**Gender**												
**Male**	1			1								
**Female**	0.578	0.280–1.194	0.139	0.601	0.282–1.282	0.188						
**Age**												
**<60**	1			1								
**≥60**	1.019	0.479–2.167	0.961	0.946	0.452–1.981	0.884						
**Smoking**												
**No**	1			1								
**Yes**	1.412	0.675–2.952	0.359	0.533	0.210–1.456	0.231						
**T stage**												
**I–II**	1			1								
**III–IV**	1.299	0.636–2.652	0.473	0.805	0.378–1.716	0.575						
**Lymph node metastasis**												
**No**	1			1								
**Yes**	3.614	1.661–7.861	0.001 *	1.391	0.684–2.829	0.363	2.198	0.945–5.116	0.068			
**Chemoradiotherapy**												
**No**	1			1								
**Yes**	1.798	0.886–3.647	0.104	1.218	0.600–2.473	0.585						
**Differentiation**												
**Well**	1			1								
**Moderate/poor**	1.196	0.459–3.120	0.714	1.374	0.479–3.940	0.554						
**DOI**												
**<5 mm**	1			1								
**≥5 mm**	4.045	1.653–9.896	0.002 *	2.236	1.039–4.816	0.040 *	1.902	0.692–5.225	0.213	2.087	0.879–4.951	0.095
**WPOI**												
**I–III**	1			1								
**IV–V**	4.311	1.851–10.040	0.001 *	1.797	0.874–3.693	0.111	1.599	0.595–4.300	0.352			
**NNMT in TCs**												
**Low**	1			1								
**High**	3.234	1.546–6.764	0.002 *	3.051	1.452–6.413	0.003 *	1.960	0.908–4.229	0.086	2.896	1.314–6.379	0.008 *

HR, hazard ratio; CI, confidence interval; * represented differences were considered statistically significant with *p* < 0.05.

**Table 3 biomolecules-12-01487-t003:** Cox-regression analysis of DFS and MFS in OSCC patients.

Variables	Univariate Analysis						Multivariate Analysis					
	Disease-Free Survival			Metastasis Free Survival			Disease-Free Survival			Metastasis Free Survival		
	HR	95%CI	*p* Value	HR	95%CI	*p* Value	HR	95%CI	*p* Value	HR	95%CI	*p* Value
**Gender**												
**Male**	1			1								
**Female**	0.584	0.326–1.047	0.071	0.344	0.107–1.106	0.073						
**Age**												
**<60**	1			1								
**≥60**	0.849	0.476–1.515	0.579	0.612	0.212–1.768	0.365						
**Smoking**												
**No**	1			1								
**Yes**	1.000	0.527–1.896	1.000	2.076	0.715–6.032	0.179						
**T stage**												
**I–II**	1			1								
**III–IV**	1.239	0.702–2.188	0.460	2.576	0.893–7.435	0.080						
**Lymph node metastasis**												
**No**	1			1								
**Yes**	2.335	1.321–4.129	0.004 *	3.904	1.213–12.571	0.022 *	1.556	0.821–2.949	0.176			
**Chemoradiotherapy**												
**No**	1			1								
**Yes**	1.866	1.069–3.258	0.028 *	2.932	0.980–8.770	0.054	1.000	0.531–1.886	0.999			
**Differentiation**												
**Well**	1			1								
**Moderate/poor**	1.414	0.535–2.437	0.733	1.400	0.312–6.278	0.660						
**DOI**												
**<5 mm**	1			1								
**≥5 mm**	2.735	1.464–5.111	0.002 *	1.686	0.560–5.074	0.353	2.306	1.152–4.616	0.018 *			
**WPOI**												
**I–III**	1			1								
**IV–V**	2.596	1.423–4.640	0.002 *	2.245	0.745–6.765	0.151	1.344	0.657–2.750	0.419			
**NNMT in TCs**												
**Low**	1			1								
**High**	2.263	1.287–3.979	0.005 *	2.221	0.767–6.427	0.141	2.104	1.155–3.833	0.015 *			

HR, hazard ratio; CI, confidence interval; * represented differences were considered statistically significant with *p* < 0.05.

## Data Availability

Publicly available datasets were analyzed in this study. These data can be found here: https://gdc-hub.s3.us-east-1.amazonaws.com/download/TCGA-HNSC.htseq_counts.tsv.gz.

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
