# Peer review of "Identification of Biological Functions and Prognostic Value of NNMT in Oral Squamous Cell Carcinoma"

_biomolecules, 2022, doi:10.3390/biom12101487_

Round 1
Reviewer 1 Report
Overall there is no gap in knowledge for this study as the detail of how NNMT involves in HNSCC has been published by Togni et al. (2021, Biomolecules). I would suggest the author to consider to investigate the effect of using 5-Amino-1MQ to replace the shRNA and validate in PDX and syngeneic mouse model.
Reviewer 2 Report
There is a little bit of a missing thread. The last paragraph of the introduction should clearly state what the aim of the study was. Before that, the scientific or clinical problem that has not yet been solved must be clearly stated. What was the purpose of the study in the first place?
In the discussion, the thread must be picked up again and the research question formulated in the introduction must be answered. This is also missing.
And finally, it is essential to determine whether the results have clinical relevance. Will the results of the study change the diagnosis in the future? Which of the many molecular biological methods of this work could be implemented in clinical routine? Even if there is not always a direct application from basic research, it should be discussed whether this could ever happen, and how this could work, and where the benefit for the patient could possibly lie.
The citation within the text is strange. One should put spaces in front of the square brackets.
The entire text should be checked for correct punctuation.
Some linguistic improvements should also be made. I would recommend consulting a professional medical writer or translator.
Since numerous statistical tests have been carried out here, it is important to correct for multiple testing. For example, a Bonferroni correction of the p-values could be recommended.
The bibliography is incomplete. Hardly any current scientific works were cited. Literature outside China was also hardly taken into account.
Round 2
Reviewer 1 Report
The author highlighted the different between their study with Togni et al. (2021, Biomolecules) is mostly due to their group had investigated the staining of NNMT in fibroblast and TIL using IHC technique. However, the entire study of the correlation is solely based on tumor cell IHC immunereactive score. I have no idea how is it different again with Togni et al.
I would suggest if the author can show the relationship of NNMT expression from the TME, it can provide a relationship of the crosstalk between tumor cell and fibroblast in the TME to enhance tumorigenesis of HNSCC. The author needs to score perhaps combination of fibroblast and tumor cell intensity with the clinicoparameters and clinical outcome of the patient.
Apart from that, the author needs to show the evidence of using CAF to investigate the role of NNMT if the author need to pitch in this line. Otherwise, there is solely no gap and purely just repeat what Togni et al. (2021, Biomolecules) study with a larger tumor size.
Reviewer 2 Report
Thank you very much for responding to my comments and implementing my suggestions.
Author Response
Thank you for the time in reviewing and for the constructive suggestions to improve our manuscript quality.